

# Semi-Analytical Methodology for Fretting Wear Evaluation of the Pitch Bearing Raceways Under Operative and Non-Operative Periods

David Cubillas [1], Mireia Olave [1], Iñigo Llavori[2], Ibai Ulacia [2], Jon. Larrañaga [2], Aitor Zurutuza [3], Arkaitz Lopez[3]

[1]Ikerlan Technology Research Centre, Basque Research and Technology Alliance (BRTA). Pº J.M. Arizmendiarrieta, 2. 20500 Arrasate/Mondragón. Spain.
[2]Mondragon Unibertsitatea - Faculty of Engineering, Loramendi 4, 20500, Arrasate-Mondragon, Spain
[3]Laulagun Bearings, Harizti Industrialdea 201-E 20212, Olaberria, Spain.

*Correspondence to*: D. Cubillas (dcubillas@ikerlan.es)

**Abstract.** Current methodologies for the evaluation of fretting in pitch bearing raceways only consider damage induced by oscillating control movements. However, pitch bearings can remain static for long operative and non-operative periods, and previous works have shown that load fluctuations cause fretting damage due the small movements and deformations at the contact. In this work a semi-analytical methodology is presented and an analysis of the fretting phenomenon in pitch bearing raceways under both productive and non-productive periods of static pitch control is accomplished. The analysis is performed following the energy-based wear model and the calculation of a total 30 times series of 10 minutes of duration concerning winds speeds from 3 m/s to 25 m/s. As result, critical locations, and critical times, defined as the maximum allowable time in the same position before the damage is developed, are presented for the different wind speeds. Additionally, a cumulative method is proposed for simplified calculations of variable winds.

## 1 Introduction

Wind power industry has experienced an extraordinary growth in last decades due to the high demand of clean and renewable energy (Nematollahi, et al., 2016) (Rodríguez, et al., 2019). This fact has been accompanied by new advances in design, materials and manufacturing technics that finally have resulted in wind turbines of larger dimension, and consequently, larger components (Serrano, et al., 2016). This fact and the extreme operational conditions have led to complex scenarios where the methodologies for common components are not reliable. While new calculation procedures and design guidelines have been proposed (Harris, et al., 2009.) (Portugal, et al., 2017) (Houpert, 1999) (Schwack, et al., 2016) (Heras, et al., 2017) (Olave, et al., 2018) for the analysis of the reliability of the bearing raceway in terms of rolling contact fatigue and fretting damage, these techniques are supported by simplified models with high safety factors what result in conservative calculus, and low optimization (Lopez, et al., 2019). This fact has promoted the extensively usage of experiments to test the reliability of the wind turbine components under their different failure modes (Stammler, et al.) (Menck, et al., 2020) (Schwack, et al., 2021).

This technique has been applied to the real scale components, and therefore, they have provided effective and reliable results.





However, they imply high costs due to the need of real scale components, materials and equipment that are even increasing due to the continuous component growth. Under this scenario, wind sector claims for the development and validation of new methodologies in multiscale components for the final application to real scale problems (Olave, 2019).

Pitch bearings are typically large 4-point contact ball (4PCB) bearings and it aim is to link the blades to the hub while allow
pitch control to locate the blade in the optimal position to increase the turbine efficiency and to reduce the structural loads when the wind is too high (Bossanyi, 2003). Due to the pitch control strategy, these bearings are usually exposed to oscillatory movements, and eventually the amplitude of the moment can be lower enough to result in rotational fretting at the bearing raceway (Schwack, et al., 2016) as consequence of the reciprocating rolling motion of the balls (Cai, et al., 2020) and the squeeze of the lubricant out of the contact between the rolling element and the raceway. This phenomenon has been extensively
studied for different authors where analytical (Cubillas, et al., 2022), numerical  (Schwack, et al., 2018) (Fallahnezhad, et al., 2018) and experimental (Grebe, et al., 2011)(Grebe, et al., 2020) (Schwack, et al., 2021) methods have been used to study the effects of these control movements on the fretting damage. As result, it has been concluded that oscillatory movements of enough amplitude promote the relubrication of the raceway and aid to avoid fretting damage. Reason why authors have proposed the premeditated use of the control movements, called endurance runs, to avoid the damage (Stammler, et al.)

Under some circumstances, such as, pre-commissioning, low wind speed or safe operating stops, pitch control can remain static for large periods of time. Under this scenario, pitch bearing still must accommodate tilting moments exerted by the wind and the blades weight that are transmitted between the rings through the balls. As consequence, balls experience variable loads what may cause radial fretting (Zhu, et al., 2006) (Cubillas, et al., 2021), and additionally, this variable load causes small deformations at the ball and the rings contact what result in small rolling movements (Olave, et al., 2018) in the transversal
direction of the raceway that squeeze out the lubricant and, finally, they may cause rotational fretting.

Despite of the invested efforts in literature on the analysis of the pitch control moments on pitch bearing raceway fretting, the effect of the variable loads when the pitch control remains static have not been studied. In previous work, the authors has developed and validated formulations for the analysis of radial fretting (Cubillas, et al., 2021), rotational fretting (Cubillas, et al., 2022) and the combination of both (Cubillas, et al., 2022) in small bearings.

The main objective of this work is to propose a complete methodology for the analysis of pitch bearing fretting damage, and to analyse the performance of the damage under productive and non-productive periods of static pitch control, to determine the critical locations, operative conditions, and critical times (defined as the maximum allowable time in the same position before the damage is developed in the absence of lubricant). For this task, the 5MW NREL reference wind turbine has been taken as case of study, and a total 30 time series of different design load cases (DLC) of normal production (DLC 1.2) and
non-productive conditions periods (DLC 6.4) are evaluated through a wear energy-based model (61400-3, 2009). Additionally, a simplified cumulative method for variable wind speed analysis is proposed.

In this work, the wear initiation has been addressed following the energy based wear model with a threshold energy of wear activation corresponding to the initial energy required to transform the metal to a tribologically transformed structure (TTS)



(Sauger, et al., 2000) (Sauger, et al., 2000) (Fouvry, et al., 2003). However, it is important to mention that the present analysis

does not consider the effect of the lubricant on the contact which could notably increase the time of wear initiation.

**2 Methodology**

In previous work (Cubillas, et al., 2022), the authors developed and validated a complete methodology to predict fretting damage in static angular bearing subject to variable loads. In this work, this methodology is extended and adapted to the 4PCB bearing problem.

Figure 1 shows the flowchart of the complete methodology. The required inputs are the wind time series containing the value of the bearing reaction along the time $M_x(t)$, $M_y(t)$, and the geometry. The maximum value of the bearing reaction and the geometry are used to perform an design of experiments FEM simulations what allow create a surface response with the values of every ball reaction and contact angle as function of the bearing reaction, ball id ($i$=1: nb), row ($j$=1:2), and contact id ($k$=1:4), see Figure 2.


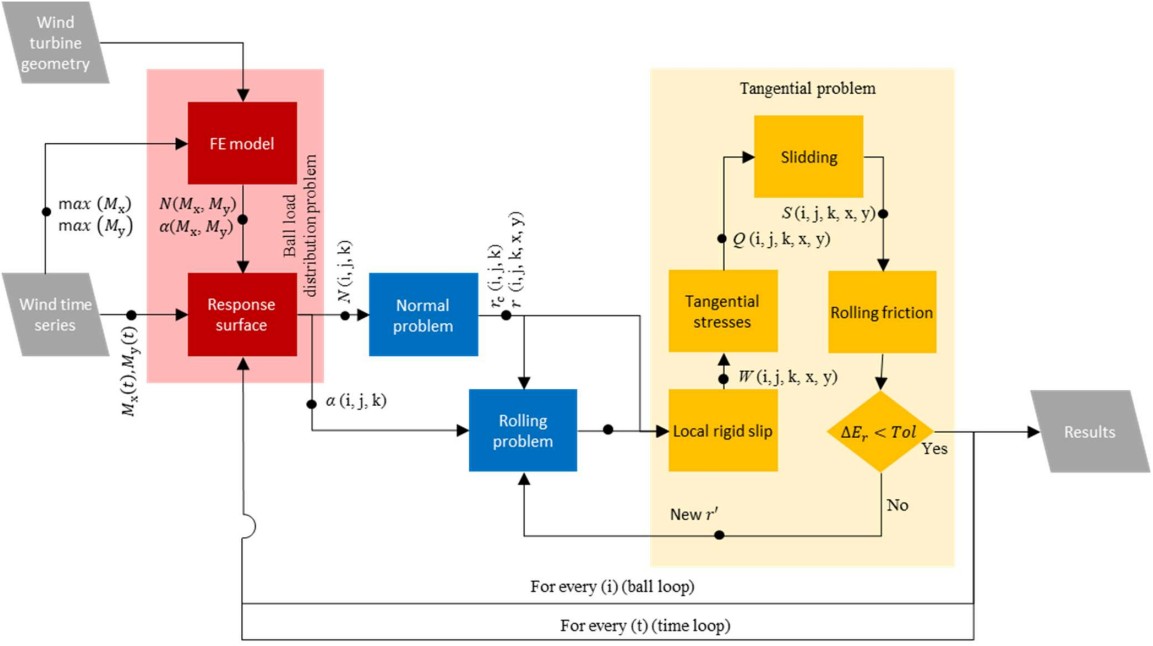

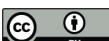



**Figure 1: Flow chart of the proposed semi-analytical formulation showing the necessary calculation process with the involved loops to calculate the fretting damage where the suffix i defines the number id of the ball, the suffix j defines the row id, the suffix k defines contact location, and the suffix t defines the time step.**

Once the response surface is created, it is possible to calculate the contact reaction, $N(t)$, and the contact angle, $\alpha(t)$, of every contact point of a ball in a specific time, then the contact reaction provides the enough information to calculate the contact deformations; and the variation in the contact angle allows to determine the ball motion. The kinematics of the ball are calculated following author´s previous work (Cubillas, et al., 2022), with an iterative solver for rolling friction minimization. This is implemented using the Nelder-Mead simplex algorithm, as described in (Lagarias, et al., 1998), with a tolerance on the

function value of 10e-4. Within this iterative solver the local rigid slip, tangential stresses, and sliding are calculated. Finally, the fretting damage and the probability of damage initiation are calculated.

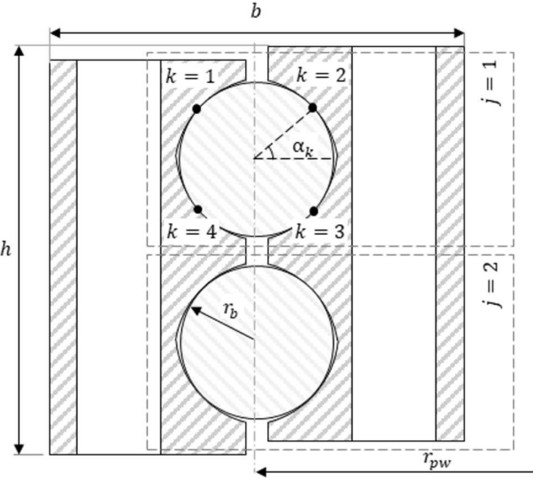

**Figure 2: Generic 4PCB bearing of main diameter $r_{pw}$, height, h, and wide, b; and the definition of the contact through the suffix j that defines the id of the row, and the suffix, k, that defines the id of the contact location.**

### 95    2.1 Wind turbine: 5MW NREL reference turbine

For the analysis purpose, the 5MW NREL case of study has been selected (Jonkman, et al., 2009). This model is a utility-scale multimegawatt turbine, conventionally three-bladed and variable blade-pitch-to-feather-controlled. It has been used as a reference turbine by research teams throughout the world to standardize baseline offshore wind turbine specifications and to quantify the benefits of advanced land- and sea-based wind energy technologies (Zuheir, et al., 2019), (Cherubini, et al., 2021),

(Halawa, et al., 2018).





Bearing case of study is a double row 4PCB bearing attached to the hub and the blade through bolts. Table 1 summarises the value of the main parameters of the bearing.

**Table 1. Bearing geometrical characterises.**

| Bearing dimensions | | | |
|---|---|---|---|
| $r_{pw}$ | Main raceway diameter | 3610.0 | [mm] |
| $r_b$ | Ball radius | 32.50 | [mm] |
| $c_O$ | Conformity | 0.52 | [ - ] |
| $D_s$ | Bolts diameter | 36.00 | [mm] |
| h | Bearing ring heigh | 200 | [mm] |
| b | Bearing ring width | 200 | [mm] |
| $n_b$ | Number of balls | 121 | [ - ] |

**2.2 Pitch control and time series**

Pitch control are usually Collective Pitch Control (CPC) or Individual Pitch Control (IPC) (Lopez, et al., 2019). While CPC defines the position of the blades simultaneously, the IPC defines the position of each blade individually. Both controls result in oscillating movements, and beside of this classification, the final routines of oscillations can be as specific as the turbine design, the world location, and the climate conditions. For this work, a reference control with 5 seeds of wind time series of the normal turbulence model (NTM) from winds of 3 m/s to 25 m/s according to the design load cases (DLC) (61400-3, 2009)

DLC1.2 (normal production) and DLC 6.4 (parked turbine with idling rotor) are available with an individual duration of 10 minutes at 20Hz. Figure 3 shows the evolution of the power production with the wind speed, and the value of the pitch bearing angle rate where the pitch control remains static for values under 7 m/s.

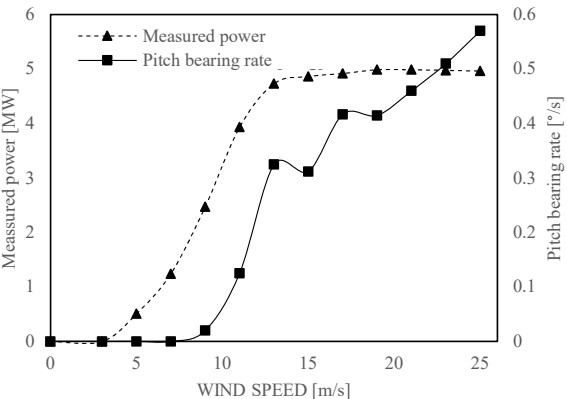

**Figure 3: Pitch control performance of the 5MW NREL reference wind turbine as function of the wind to maximum 25 m/s speed**
**and the measured power [10].**




Following above observations, only series under 7 m/s are considered from DLC 1.2, series of 3 m/s, 5 m/s, and 7 m/s, and all available series from DLC 6.4, series of 3 m/s, 11 m/s and 25 m/s. The aeroelastic wind time series are decomposed in bearing reactions using the software BLADED (DNVGL-ST-0437, 2016).

**120**  **2.3 Ball load distribution and contact angle problem**

As describe in (Cubillas, et al., 2022), a finite element (FE) model is necessary to calculate the contact reaction, $N$, and the contact angle, $\alpha$, of every contact of every ball under a specific bearing load scenario.

$$N = f\left(M_x, M_y, i, j, k\right),\tag{1}$$

$$\alpha = f\left(M_x, M_y, i, j, k\right),\tag{2}$$

However, this method is time consuming, and the calculation of every time increment of the time series is not operational. To manage this inconvenient, an experiment design is accomplished covering all possible load cases (see Table 2) what allows to
**125**  create a response surface. Radial forces and axial forces are neglected in this analysis as it has been demonstrated not to have a considerable effect on the bearing load distribution (Portugal, et al., 2017). A total 8 simulations are performed where the load is gradually applied in 10 load increments resulting in a total 80 load cases.

**Table 2. DoE for the development of a surface response of the contact reaction and contact angle.**

| Id | 1 | 2 | 3 | 4 | 5 | 6 | 7 | 8 |
|---|---|---|---|---|---|---|---|---|
| Mx | $M_x^M$ | $M_x^M$ | 0 | $-M_x^M$ | $-M_x^M$ | $-M_x^M$ | 0 | $M_x^M$ |
| My | 0 | $M_y^M$ | $M_y^M$ | $M_y^M$ | 0 | $-M_y^M$ | $-M_y^M$ | $-M_y^M$ |

**130**  The FE model is built in ANSYS considering not only the bearing geometry but also the geometry of the blade, bolts, and stiffeners. Nevertheless, a symmetric behaviour is assumed and only one blade and 1/3 of the hub is considered, see Figure 4a. Bearing, and stiffeners are structural steel, hub is cast iron and blade is laminate. Table 3 summarizes the elastic properties of the mentioned materials.

**Table 3. Elastic properties of the wind turbine components**

| Property | Cast iron | Struct. steel | Lam. | Units |
|---|---|---|---|---|
| Elastic module | 210 | 110 | 40 | [GPa] |
| Shear module | 76.9 | 42.9 | 15 | [GPa] |
| Poisson ratio | 0.3 | 0.28 | 0.3 | [ ] |

**135**





Mesh is covered with second order elements and a total 1562761 elements are used in the entire model with a maximum value of the element size of 10 mm in the bearing. Balls are replaced by two linear elements with variable stiffness connected to both bearing raceways (Daidié, et al., 2008), and analogously, bolts are replaced by linear elements considering the minimum section see Figure 4b.

Moment is applied through a remote point located at bearing centre and linked to the blade with a flexible behaviour. In this way, the decomposition of the moment is prevented, and a true moment is ensured. Attending to the previous assumption of symmetric behaviour of the hub, frictionless supports are applied to the cut surfaces to prevent normal deformations and allow deformations along the symmetry plane. Frictional contacts are used at the contacts between the bearing rings, stiffeners, and blade with a 0.3 coefficient of friction with asymmetric behaviour and augmented LaGrange formulation. Bolts extremes are

connected to the bearing rings, blade, and hubs with a MPC contact, and a 608 kN of pretension is applied to the bolts based on a 10.9 grade.

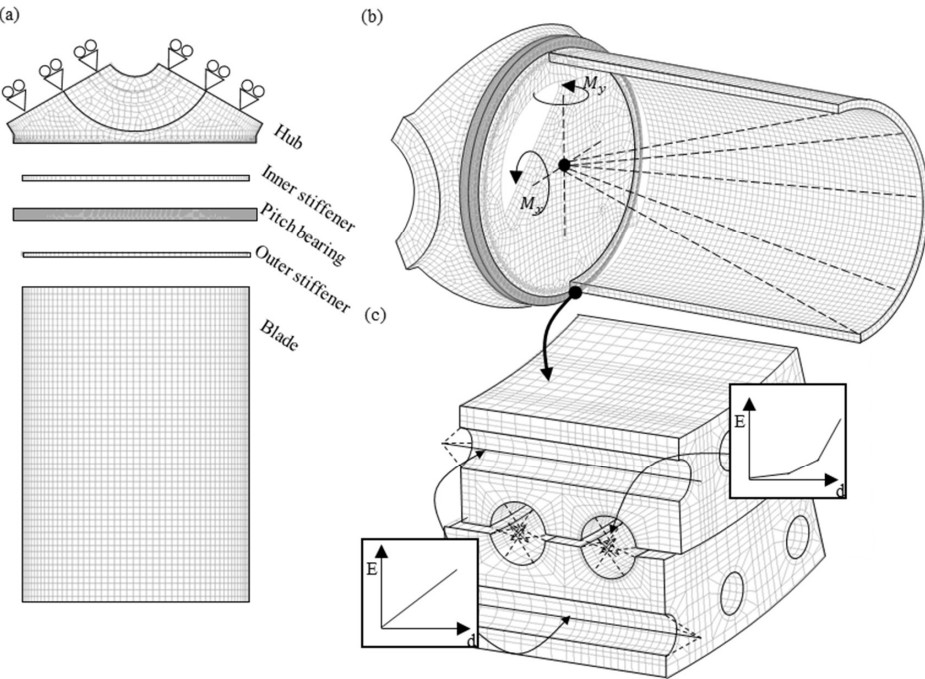

**Figure 4: FEM model of the 5MW NREL reference wind turbine formed by the hub, the pitch bearing, the blade, and the inner and outer stiffeners; (a) Simplification of the original geometry considering a symmetric behaviour of the hub; (b) Application of the**
**tilting moments through a remote point located at the bearing centre;(c) Detail of the bearing mesh and the simplifications of balls and bolts by a spring with non-linear and linear behaviour respectively.**



### 2.4 Contact problem

Brewe and Hamrock formulation (Brewe, et al., 1977) can be used to define contact area through the mayor and minor ellipse axis, $a$ and $b$, respectively (see Figure 5a):

$$a = a^* \cdot \left( \frac{3 \cdot N_{ij} \cdot (1 - v^2)}{S_{ij} \cdot E} \right)^{1/3}, \tag{3}$$

$$b = b^* \cdot \left( \frac{3 \cdot N \cdot (1 - v^2)}{S_\rho \cdot E} \right)^{1/3}, \tag{4}$$

where, E is the elastic modulus of the material, $v$ is the Poisson´s ratio, $S_\rho$ is the sum of the contact curvatures and $a^*$ and $b^*$ are the dimensionless semiaxes quantity of the mayor and minor semiaxes respectively. APPENDIX A describes the calculation of the above variables.

Additionally, both bearing raceway and ball deform to finally coincide in a transversal curvature (Hertz, 1896), $r_c$:

$$r_c = \frac{2 \cdot r_b}{1 - s} \tag{5}$$

### 2.5 Transversal rolling problem

Following our previous work (Cubillas, et al., 2022), the travelled distance of the ball through the raceway, $\Delta x$, (see Figure 5a) can be calculated as:

$$\Delta x = \Delta \alpha \cdot r_c, \tag{6}$$

and the creepage, $\varepsilon$, is calculated as:

$$\varepsilon_x = \frac{r - r_e}{r_e}, \tag{7}$$

where, $r_e$ is the effective radius (calculated in the iterative process of energy minimization) and $r$ is the distance from any point to the rolling axis:

$$r = \sqrt{r_c^2 - x'^2} - \sqrt{r_c^2 - a(y')^2} + \sqrt{r_b^2 - a(y')^2}, \tag{8}$$

where, x' and y' are the local coordinates of the ball according to Figure 5a, $a(y)$ is the distance from a point to the ellipse bound in x direction, and $b(x)$ is the distance from a point to the ellipse bound in y' direction. Both can be calculated respectively as:

$$a(y') = \sqrt{a^2 \cdot \left( 1 - \frac{y'^2}{b^2} \right)}, \tag{9}$$

$$b(x') = \sqrt{b^2 \cdot \left( 1 - \frac{x'^2}{a^2} \right)}, \tag{10}$$





### 2.6 Tangential friction problem

Following Kalker (Kalker, 1981), the relative motion of two points, $\vec{S}$ (local slip), can be decomposed in the rigid

displacements, $\vec{W}$ (local rigid slip), and the elastic displacements, $\vec{U}$ (local elastic slip):

$$\vec{S} = \vec{W} - \vec{U}, \tag{11}$$

Considering an initial condition of no slip, then, the local elastic slip is equal to the local rigid slip:

$$\vec{S} = 0 \rightarrow \vec{U} = \vec{W}, \tag{12}$$

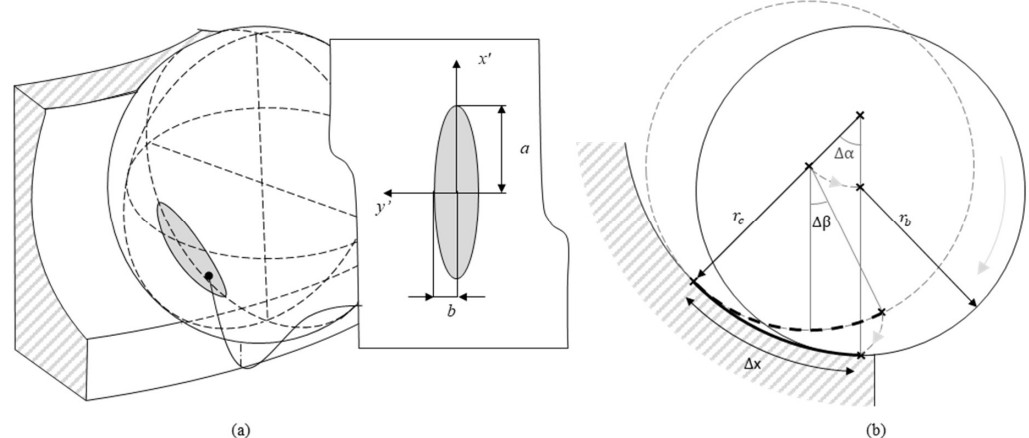

**Figure 5: Contact area and rolling motion of the ball. (a) Definition of the elliptical contact area through the mayor ellipse semiaxes a, and the minor semiaxes, b, and the local coordinates y' aligned with the longitudinal direction of the raceway and x' aligned with**

**the transversal direction of the raceway; (b) description of the rolling motion of the ball in the raceway transversal direction**

Following step is to determine the local rigid motion at the contacting points:

$$\vec{W}(W_{x\prime}, W_{y\prime}) = W_{i,j,k}^{NL}(W_x^{NL}, W_y^{NL}) + W_{i,j,k}^{Ro}(W_x^{Ro}, W_y^{Ro}), \tag{13}$$

where, $W^{NL}$ is the local rigid slip caused by the variable normal load, $W^{Ro}$ is the local rigid slip caused by the rolling effects.

According to our previous work (Cubillas, et al., 2021), the local rigid slip caused by the normal load is calculated as:

$$W_x^{NL} = r_c \left( \tan^{-1}\left(\frac{x'}{\sqrt{r_p^2 - x'^2}}\right) \frac{\tan^{-1}\left(\frac{a(y')}{\sqrt{r_c^2 - a(y')^2}}\right)}{\tan^{-1}\left(\frac{a(y')}{\sqrt{r_p^2 - a(y')^2}}\right)} - \tan^{-1}\left(\frac{x'}{\sqrt{r_b^2 - x^2}}\right) \frac{\tan^{-1}\left(\frac{a(y')}{\sqrt{r_c^2 - a(y')^2}}\right)}{\tan^{-1}\left(\frac{a(y)}{\sqrt{r_b^2 - a(y')^2}}\right)} \right) \tag{14}$$



$$W_y^{NL} = r_c \left( \tan^{-1}\left(\frac{y'}{\sqrt{r_b^2 - y'^2}}\right) \frac{\tan^{-1}\left(\frac{b(x')}{\sqrt{r_c^2 - b(x')^2}}\right)}{\tan^{-1}\left(\frac{b(x)}{\sqrt{r_b^2 - b(x')^2}}\right)} - y \frac{\tan^{-1}\left(\frac{b(x')}{\sqrt{r_c^2 - b(x')^2}}\right)}{b(x)} \right),$$ (15)

On the other hand, the local rigid slip caused by the rolling motion considering the transient phenomena (Al-Bender, 2008)

can be calculated as:

If $a(y') - \Delta x \leq x$ and $x < a(y')$:

$$Wx_{Rx} = \varepsilon \cdot (x' - a(y'))$$ (16)

If $a(y') - \Delta x' > x'$ and $x' \geq -a(y')$:

$$Wx_{Rx} = \varepsilon \cdot \Delta x'$$ (17)

**2.7 Tangential stresses, sliding and rolling energy**

Kalker simplified theory (Kalker, 1982) is applied to determine the tangential stress $\vec{Q}$:

$$\vec{Q}(x', y') = \frac{\vec{W}(x', y')}{L},$$ (18)

where, L is the flexibility parameter calculated as per APPENDIX B.

However, tangential stresses must be reconsidered at those points where slip occurs, if $Q(x, y) > \mu \cdot P(x, y)$:

$$Q(x', y') = \mu \cdot P(x', y')$$ (19)

$$U(x', y') = \mu \cdot P(x', y') \cdot L$$ (20)

where, $\mu$ is the value of the coefficient of friction (CoF), and $P$ is the contact pressure:

$$P(x', y') = \frac{3 \cdot N}{2\pi \cdot a \cdot b} \cdot \sqrt{1 - \left(\frac{x'}{a}\right)^2 - \left(\frac{y'}{b}\right)^2}$$ (21)

then, reconsidering (11), the local slip:

$$S(x, y) = W(x, y) - U(x, y)$$ (22)

Finally, the rolling energy, $E_R$ is calculated from the local terms:

$$E_R = \iint S \cdot Q_l + U \cdot Q \, dxdy$$ (23)

**2.8 Fretting damage indicators**

An energy based model has proposed and validated (Fouvry, et al., 2003) for fretting applications, considering the dissipated friction energy where the wear volume, $V_w$, is proportional the total accumulated friction energy (TFE), $E_F$ and the wear coefficient, $k_w$:

$$V_w = k_w \cdot E_F,$$ (24)

where $E_F$ is calculated from local terms as:





$$E_{\mathrm{F}} = \iint S(x',y') \cdot Q(x',y') \cdot dxdy. \tag{25}$$

The dissipated energy can be also expressed as a distribution through the density of the accumulated friction energy (DFE), $\rho_{\mathrm{F}}$:

$$\rho_{\mathrm{F}}(x',y') = S(x',y') \cdot Q(x',y') \tag{26}$$

Following this work, multiple authors have found analogues result in multiple fretting wear applications, and in particular for false brinelling prediction in roller bearings (Fallahnezhad, et al., 2019), (Fallahnezhad, et al., 2018), (Brinji, et al., 2021), angular bearings (Schwack, et al., 2018), and thrust bearings (Cubillas, et al., 2022), (Cubillas, et al., 2021). However, the

value of $k_w$ has shown dissimilar results as it is affected by multiple conditions such as frequency, sliding distance or pressure (Brinji, et al., 2021).

The energy-based model also suggests the existence of a threshold energy for wear activation, $k_A$, which is a initial input of energy to form the tribological transformed structure (TTS) before starting wear (Sauger, et al., 2000). In recent work (Brinji, et al., 2021), it has accomplished the characterization of the behaviour of high carbon bearing steel under varying conditions

for false brinelling applications where it has been concluded that the value of the threshold energy for wear activation is not affected by altering external conditions of normal pressure, oscillating sliding frequency or sliding amplitude, and can be approximated to 1 Julee, which is also close to similar applications (Sauger, et al., 2000), (Pearson, et al., 2014), (Ramalho, et al., 2006).

**2.9 Probability of damage initiation**

The accumulated friction energy is expected to growth during the entire duration of the analysis, and therefore a rate of dissipated friction energy, $\dot{E}_{\mathrm{F}}$, can be defined as the ratio between accumulated friction energy and the duration of the time series, $t$:

$$\dot{E}_{\mathrm{F}} = \frac{dE_{\mathrm{F}}(t)}{dt}, \tag{27}$$

Due to the random loading from the wind, this growth is not expected to be linear with the time. However, a mean value, $\Delta E_F$, can be calculated as the accumulated friction energy during the total simulation and the duration, $t_s$.

$$\Delta E_F = \frac{E_{\mathrm{F}}}{t_s}, \tag{28}$$

As previously described, the damage initiates when $E_{\mathrm{F}}$ achieved the value $k_A$. Then, a critical time, $t_c$, defined as the maximum allowable time in the same position before the damage is developed, can be estimate:

$$t_c = \frac{k_A}{\Delta E_F}, \tag{29}$$

A total 5 seeds of each wind speed (3 m/s, 5 m/s and 7 m/s) are performed; therefore, 5 different critical times are expected for the same wind speed. Under this scenario, the mean value, $\mu_{tc}$, and the typical deviation, $\sigma_{tc}$, are calculated:





$$\mu_{tc} = \frac{\sum t_{c,i}}{n_s}, \tag{30}$$

$$\sigma_{tc} = \sqrt{\frac{\sum (t_{c,i} - \mu_{tc})^2}{N_s}}, \tag{31}$$

where, $n_s$ is the number of simulations.

Finally, the density of normal probability distribution of damage initiation can be reached,

$$X = \frac{e^{\left(\frac{t-\mu_{tc}}{2\cdot\sigma_{tc}^2}\right)}}{\sigma_{tc}\cdot\sqrt{2}}, \tag{32}$$

and the function of the distribution of normal probability, $\Phi_X$,

$$\Phi_X = \frac{e^{\frac{t-\mu_{tc}}{2\cdot\sigma_{tc}^2}}}{\sigma_{tc}\cdot\sqrt{2}} \tag{33}$$

## 3 Results and discussion

In this section the results from the simulations are presented. First, the load and contact angle distributions are studied to
determine the areas that are exposed the most to variable loads and variable contact angles. Then, the distribution of the damage
is analysed and compared to the load distribution to determine its effects on the final damage, and the effects of the wind;
finally, the results of the probability of damage initiation are presented, and a cumulative model for variable winds analysis is
proposed.

### 3.1 Load distribution and contact angles

Figure 6 shows the distribution of the mean and the amplitude load, the mean and the amplitude contact angle and the damage
indicators, the total accumulated friction energy, TFE, and the maximum density of friction energy, MDFE, after the simulation
of the total 15 wind series of 10 minutes of the DLC 1.2 and 6.4 respectively. For a clearer presentation of the results, contacts
are classified attending to the load diagonal, the row, and the ring (inner and outer), and attending to the high number of
contacting points and the similarity between adjacent locations, the results correspond to the mean value of two adjoining balls
with the objective of facilitate the presentation of the data.

In this section, we first focus on the analysis of the loads and the contacts angles. The results reveal the presence of variable
loads and rolling motion (as consequence of the variable contact angle), and therefore the potential presence of radial and
rotational fretting damage. Previous classification into load diagonals, rows, and rings, allows to easily observe a same value
of the mean and amplitude load and mean and amplitude contact angle at the inner and the outer rings as the contact reactions
are equal in magnitude with opposite direction.





**Figure 6:** Summary of the results of the analysis of the mean load, amplitude load, mean angle, amplitude angle, the total dissipated friction energy (TFE) and the maximum value of the density of accumulated friction energy (MDFE) where results are classified attending to the row location, the load diagonal and for the different design load cases DLC 1.2 and 6.4




The load and the contact angle show a symmetric behaviour respect to a crossing axis from 150º to 330º (see back dashed line in Figure 6). Additionally, the values of the mean load and the mean angle on one side, and the amplitude load and the amplitude angle on the other side, show a visible correlation. However, notable differences can be observed as the maximum values are not always located at the same angular position. Another interesting point is the dissimilar distribution between rows that can be observed through the maximum values at the row 1, and row 2, being significantly different. These facts show the important effect of the heterogeneous stiffness of the structure and endorse the usage of the FE model to calculate the bearing reactions as shown in (Olave, et al., 2018). The distribution of the mean load with highest values located at D1 at 150◦, and D2 at 315◦ seems to reveal the presence of a permanent tilting moment of similar components $M_x$ and $M_y$ that might be attributed to the mean load exerted by wind. The highest values of the mean load are 30 kN for DLC 1.2 and 12 kN for DLC 6.4 at D1R2 and D2R1. Similarly, higher values of the mean angle can be found in close locations at 120◦ in D1 and at 320◦ in D2 with similar maximum mean values of 58◦ at D2R1 and D1R2 for the DLC 1.2 and 50◦ for the DLC 6.4. On the other hand, the results of the analysis of the DLC 6.4 show a more homogeneous distributed load over the bearing, where the lower value of the tilting moment might amplify the effects of the heterogeneous stiffness resulting in different locations of the maximum values of the loads and the contact angles.

On the other hand, the higher values of the amplitude load are located at 60º, 150º and 225º in D1, and 30º and 270º in D2 with a maximum value of 13kN for the DLC 1.2 and 4 kN for the DLC 6.4. This distribution reveals the presence of a complex scenario with an alternative tilting moment of components Mx and My caused by the alternative load of the blade weigh due to the turbine movement, and the fluctuation of the wind. The regions of high variable load reveal critical regions for the occurrence of radial fretting. Similarly, the higher values of the amplitude of the contact angle can be found at 90◦ and 225º at D1, and 20 º and 300º at D2 with a maximum value of 6 º for the DLC 1.2 and 1.9º for the DLC 6.4 at D2R1. This regions of high amplitude of the contact angle indicates the presence of rolling motion of the balls, and consequently, they could be critical for the development of rotational fretting. The amplitude of the contact angle indicates the presence of rolling motion of the balls, and consequently, these regions are critical for the development of rotational fretting.

### 4.2 Damage indicators

In Figure 6, both damage indicators, the total accumulated friction energy, TFE and maximum density of friction energy, MDFE, show similar distribution of the damage being the TFE maximum at 60º, 135º, and 180º at D1, and at 30º and 275º in D2, and the MDFE at 275 in D2R1 for the DLC 1.2; and at 290º in D2R1 for the DLC 6.4. These damage distributions keep certain similarities with the mean and amplitude load distributions and the mean and amplitude angle pointed in the previous section. However, the correlation of the distribution of damage seems to hold the best fit with the distribution of the amplitude load. Therefore, results might indicate that the main source of damage is the radial fretting caused by the variable load of the wind and the blade weight dynamics. This fact is aligned with previous work (Cubillas, et al., 2022) where the analysis of an angular bearing indicated that the 70% of the fretting damage caused for a variable load was produced by radial fretting. In addition, the TFE and the maximum value of the MDFE show damage distributions where the inner ring is significantly more




damaged in all diagonals and rows. The value of TFE in the IR is from 1.8 to 4.6 times higher than the OR, and similarly, the value of the MDFE of the IR is from 1.2 to 3.6 times the OR. As described in previous section, the value of the contact reaction and the contact angle are equal in magnitude for the inner and the outer ring contact, therefore, no differences can be made in

terms of load or motion, and consequently, the different damage can only be attributed to the different contact curvatures of the bearing raceway where the inner contact is convex respect to the longitudinal curvature ($r_b$ vs $r_{pw}$), and concave respect to the transversal direction ($r_b$ vs $r_p$), while the outer ring contact is concave respect to both transversal and longitudinal curvatures. Therefore, the increasing main diameter of the bearings would aid to balance the damage at the inner ring.

Figure 7 shows the density of the friction energy, DFE, corresponding to the location of the maximum values at D2R1 from

225º to 320º denoted by the regions a and b for the DLC 1.2, and c and d for the DLC 6.4 (see Figure 6). At this point is important to mention that the distance between marks was reduced to facilitate the comparison between the adjacent damages and to fit the higher number. The results show notable differences in the damage shapes for every contact where some look similar to the previous experimental results (Cubillas, et al., 2022). All marks can be bounded in an elliptical area which is notable more rounded at the OR and more flattened at the IR what, as previous mentioned, is directly influenced by the contact

curvatures of the rings.

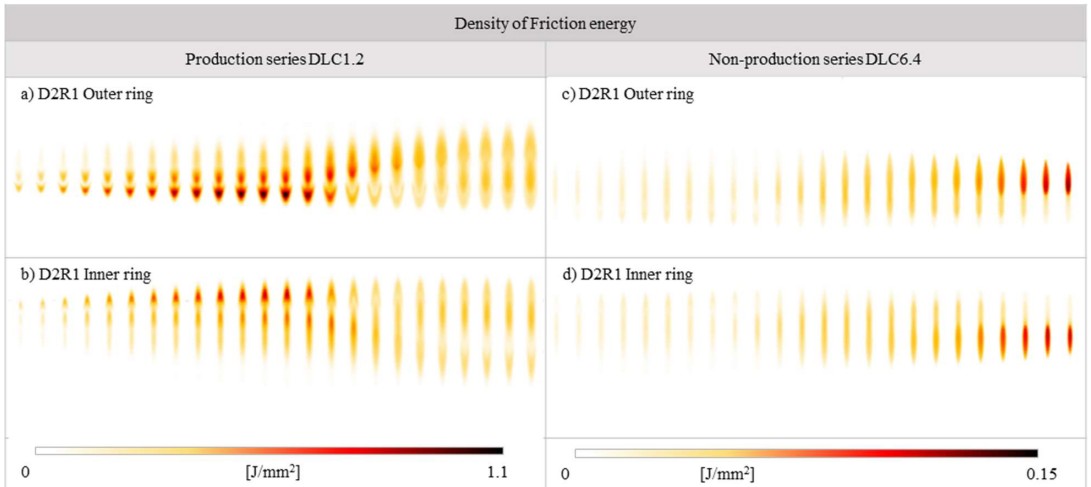

**Figure 7. Density of dissipated friction energy, DFE, at the location of its maximum values defined in Fig 7.**

### 4.3 On the effect of the wind speed

Figure 8 shows the results of the total accumulated friction energy, TFE, for different wind speeds: 3 m/s, 5 m/s and 7 m/s of

the DLC 1.2 and 3m/s 11m/s and 25 m/s of the DLC 6.4. Due to the similarities of the dagame distribution for both indicators, the TFE and the maximum density of friction energy, MDFE, only the results of one of them, the TFE, is presented, and following previous sections, the results are classified by the load diagonal, row, and ring.





Results show a similar distribution of the dissipated energy where the location of the damage seems not to be affected by the wind speed and the maximum values can be found at 60º, 135º, and 190º at D1, and at 30º and 270º in D2. However, a notable

effect is observed on the value of the TFE where the results show a growing tendency of the dissipated energy with the increasing wind speed. Figure 9 shows the normalized value of the maximum TFE and the MDFE for the different wind speeds. Despite of the analogous growing tendency on both indicators with the growing wind speed, the tendency is significantly different: while the TEF has an exponential growth, the MDFE shows a logarithmic tendency.



**Figure 8: Results of the total dissipated friction energy for different wind speeds of the design load cases DLC12 and DLC64 of 10 minutes of duration, where results are organized by load diagonal and row.**

**4.4 On the probability of damage initiation**

Figure 10 shows the evolution over the time of the total accumulated friction energy TFE produced by a ball located in D2R1 at 275º under of the DLC 12 corresponding to the wind speeds 3 m/s, 5 m/s and 7 m/s. The results show a continuous growing

tendency which is close to a linear tendency. This fact may aid to simplify the complex phenomena of the fretting scenario and endorse the usage of a mean rate of friction energy dissipation.



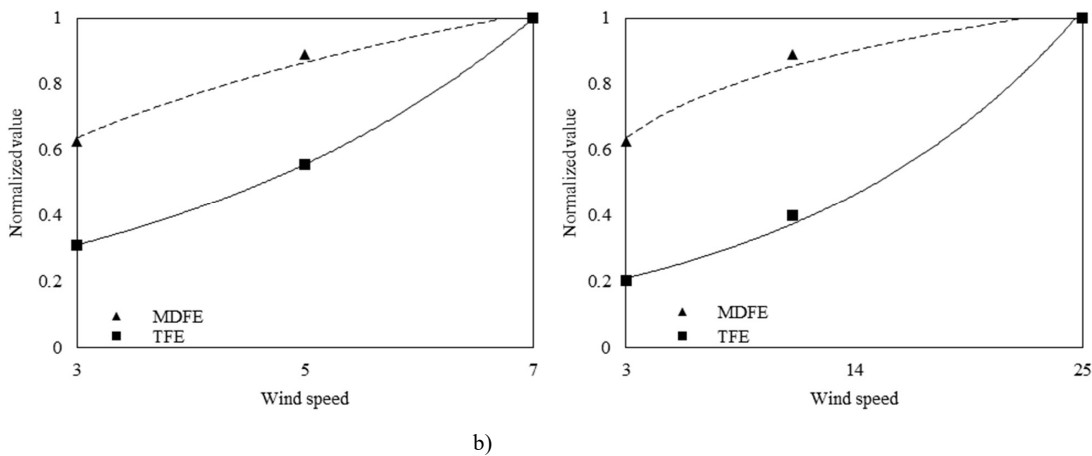

a)                                                                    b)

**Figure 9: Evolution of the normalized values of the maximum density of friction energy MDFE and the total dissipated friction**
**energy TFE as function of the wind speed, and the logarithmic and exponential regressions, respectively. a) Design load cases DLC**
**1.2; (b) Design load cases DLC 6.4.**

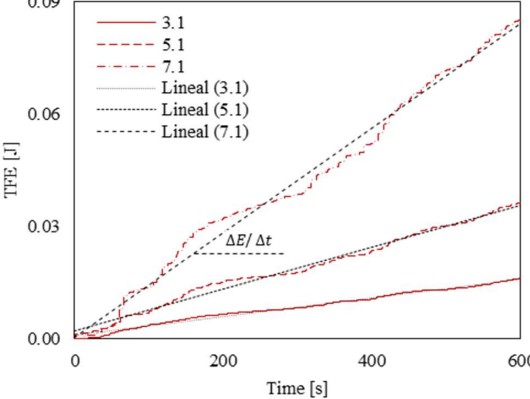

**Figure 10: Evolution of the total dissipated friction energy of the ball 105 as function of the time and for different wind speeds and**
**the corresponding linear regressions.**

Figure 11a and b summarizes the results of rate of the total accumulated friction energy, Δ(TFE), the critical time, the mean

critical time, and the deviation of the critical time for all the simulations. As mentioned, the rate of TFE grows with the wind

speed, and consequently, the critical time decreases from 4.8 hours (263.6 minutes) at 3 m/s to 1.36 hours (82 min) at 7 m/s

for production time series (DLC 1.2) and from 68.18 days at 3 m/s to 13 days at 25 m/s for the non-production time series

(DLC 6.4).

Figure 12a and b show the normal distribution of the probability of damage initiation as function of the time and for different

wind speeds of the DLC 1.2 and DLC 6.4 respectively. Figure 12a shows a fast transition between of the probability for all





wind speeds where the probability grows from the 1% to the 99% in only 20, 26 and 20 minutes for the winds of 3, 5 and 7 m/s, respectively. On the other hand, Figure 12b shows a slow transition due to the high deviation of the results of the DLC 6.4.

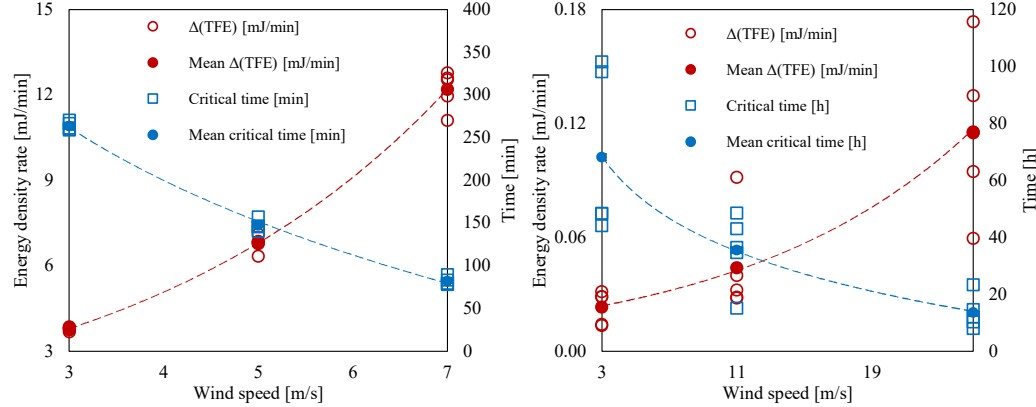


**Figure 11: Rate of the accumulated energy, critical time, mean critical time and deviation at different wind speeds and operative conditions. a) Rate of the accumulated energy, critical time, mean critical time and deviation for the DLC 1.2 from 3m/s to 7m/s; (b) Rate of the accumulated energy, critical time, mean critical time and deviation for the DLC 6.4 from 3m/s to 25m/s.**

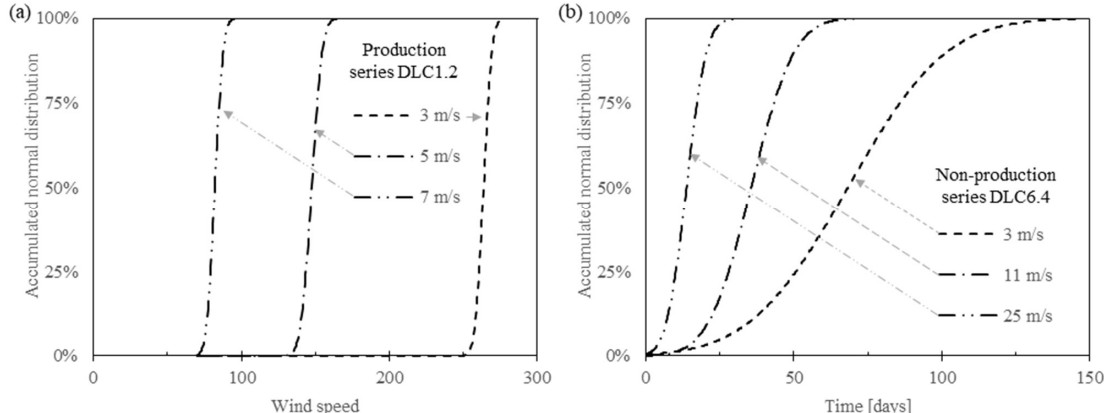

**Figure 12: Distribution of the accumulated probability of the wear initiation according to the critical time obtained with the wear activation energy of 1J; (a) Results of the analysis of the design load case DLC 1.2; (b) Results of the analysis of the design load case DLC 6.4.**

**4.5 A proposal of a cumulative rule of wear activation for variable wind**

The analysis of the critical time in previous section has resulted in different distributions of probability for each wind speed.

However, considering the real operating conditions where the wind speed is not discrete (it can take intermediate values) and



is also variable, a calculation method that allow to determine the critical time for any value of the wind speed, and to accumulate the damage in order to know the condition state under a variable wind.

During the analysis of the wind speed effect of the previous section, an exponential behaviour of the TFE was observed (see Figure 9). Therefore, the critical time associated with the same probability is expected to follow a logarithmic trend as function

of speed. This can be observed in Figure 13a and b where the logarithmic regression of the 16%, 50%, 84% and 99% probability lines are shown as a function of the wind speed for DLC 1.2 and 6.4 respectively.

Another observed characteristic was the linear behaviour of the rate of energy dissipation, see Figure 10. Consequently, a constant value of the dissipation rate could be assumed and, thus, a proportionality between the exposed time, t, the critical time, $t_c$, the activation energy, $E_a$, and the energy accumulated during that period, $E_d$.

$$\frac{E_d}{E_a} = \frac{t_i}{t_c^i} \tag{34}$$

Then, considering a number of events of different wind speed, $K$, associated with a different duration, $t_i$, and a critical time $T_c^i$, a cumulative model could be used:

$$\sum_i^K \frac{t_i}{T_c^i} \leq 1 \tag{35}$$

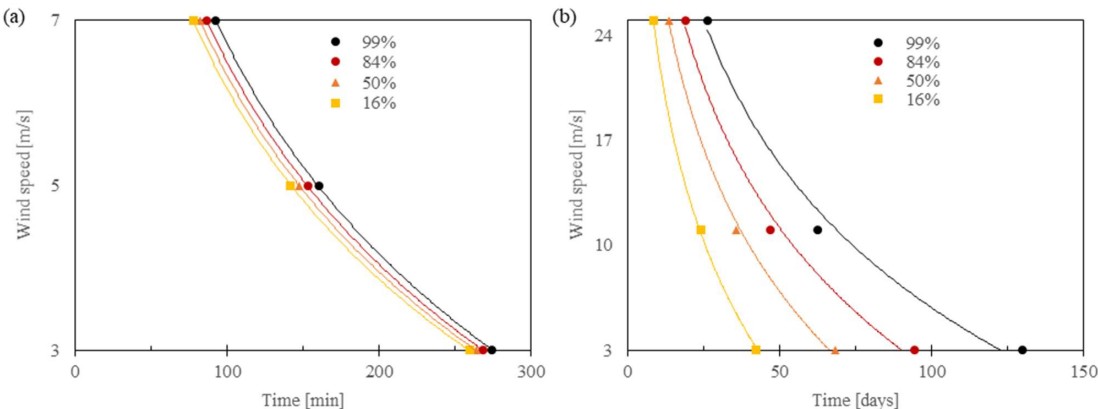

**Figure 13. Lines of probability of wear initiation as function of the wind speed and the static time of the pitch assuming a linear**
**behaviour of the rate of dissipated friction energy and taking a logarithmic regression.**

**5 Conclusions**

This work started with the aim to analyse the performance of fretting damage on productive and non-productive periods of pitch bearings under realistic time series and determine the critical locations, the conditions that favoured the development of



the damaged, and the critical times for the initiation of the damage based on the activation energy. As result, following

conclusions have been taken:

- The energy-based wear model shows a critical region between 270º to 315º of the inner ring and at the first row where the total accumulated friction energy (TFE) and the maximum value of the density of friction energy (MDFE) are located under both, operational and no operational conditions.

- An analysis of the loads and contact angle distribution shows a noticeable correlation between the friction energy and

365       the variable load, and consequently, it could imply that radial fretting is the main source of damage.

- The evaluation of the effects of the wind speed on the damage shows a similar distribution of the damage where maximum values remain at its position. However, important differences are found in the value of the accumulated energy indicators that show an exponential behaviour of the TFE, and a logarithm trend of the MDFE as function of the wind speed.

- The results of the critical time for a wear initiation based on the activation energy reveals a quick and concerning mechanism of damage in productive periods that can developed damage after 80 minutes in the worst scenario.

- Considering the close to linear trend of the rate of accumulated energy, a cumulative rule for the prediction of the wear initiation of variable speeds of wind was proposed.

### Acknowledgments

This work was partially supported by grant 20AFW2201900007 of the Bikaintek Program for the Completion of Industrial Doctorates and for the Incorporation of Researchers to the Industry from the Basque Government's Department of Economic Development, Sustainability and Environment.

The authors gratefully acknowledge the financial support given by the Eusko Jaurlaritza under 'Programa de apoyo a la investigación colaborativa en áreas estratégicas' (Project BISUM: Ref. KK-2021/00089).

### 380  Competing interests

The contact author has declared that none of the authors has any competing interests.

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
