# Peer review of "Semi-Analytical Methodology for Fretting Wear Evaluation of Unlubricated Pitch Bearing Raceways Under Operative and Non-Operative Periods"

_Wind Energy Science, 2024_

## Author Response (AR1)

**ANSWERS TO RC1**:

The authors describe a method to predict initiation of wear damage in unlubricated pitch bearings, both for the case of oscillatory movements and rotatory standstill under varying loads. The method draws upon previous works by the authors. A verification or validation by test is not part of the work.

General comments:

1- RC: Neglecting the influence of lubricant makes this model useless in the commercial design of pitch bearings. It is still an impressive work and shows the potential to become a valid and esteemed design tool one day, but right now it is not. I would thus suggest to point to the absence of lubricant in the title of the work. This will give you both the potential to highlight future works with consideration of the lubricant and also not mislead the readers.

A: We fully agree that the effect of the lubricant can have a high influence on the result, and the prediction of critical times and wear is not ready for industrial use. Also, we are very pleased to read that you see potential in the tool for further work.

Following the comments received by both reviewers, we have deeply reflected on the scope of the paper and the validated frame of the method, and have concluded that we may have overextended the capacity of our method, especially to obtain the wear and critical times. As mentioned, these are highly affected by the behavior of the lubricant, and without further methodological development the reliability of the results is low.

For this reason, and keeping the initial objective of the paper ("to propose a complete methodology for the analysis of pitch bearing fretting damage, and to analyze the performance of the damage under productive and non-productive periods of static pitch control, to determine the critical locations, and assess the worst working conditions") we have decided to eliminate the critical time calculation sections and we have focused on the analysis of critical zones and effects of wind speed and operating conditions, extending the discussion to draw more conclusions, where the ability of the energy dissipation indicator to detect critical zones and to compare results is well proven in literature.

In any case, it seems important to us to clarify to the reader from the beginning the lack of additional formulation to model the lubricated contact in boundary regime, so we have decided to incorporate it explicitly in the title and introduction.

New title:

"Semi-Analytical Methodology for Fretting Wear Evaluation of Unlubricated Pitch Bearing Raceways Under Operative and Non-Operative Periods"

Introduction:

"It is important to mention that the present analysis does not consider the effect of the lubricant in the formulation of the method. However, considering the boundary lubrication regime condition of this case of study, as well as previous work where an adequate correlation between friction energy density intensity and damage has been found, it is considered that the proposed method generates a framework of confidence for the prediction of the most critical areas where damage can occur, as well as providing a basis for comparison between different wind speeds and operating conditions."

2- A work with this amount of formulas needs a nomenclature. I would also highly recommend to follow the nomenclature of the new DG03 (nrel.gov/docs/fy24osti/89161.pdf) to make it easier for pitch bearing designers to follow your thoughts. This new DG03 got published a few days after you uploaded the first version of the paper, but I would really recommend to consider it in your literature instead of the old version.
R: We totally agree that a nomenclature list should be included. We have added it, and following your advice we have reviewed and updated it to make the variables consistent with the new DG03 guideline.

3- I would also highly recommend to consider de la Presilla et al.'s review on oscillating tests https://doi.org/10.1016/j.triboint.2023.108805 . It gives a neat and consistent definition of damage modes in oscillating bearings which you could consider. I think relying solely on the term fretting is a shortcoming and does not cover the phenomena happening under the conditions you describe. In case of doubt, I would recommend to get in touch with Markus Grebe (you cited him) who is a true expert in this field and can give you advise on the wording. Mentioning above review ( https://doi.org/10.1016/j.triboint.2023.108805 ): The works of Pitroff and Lin (2022) are related to yours and should appear in the introduction. In terms of standstill tests for pitch bearings (though for rollers), the following reference might be considered: https://doi.org/10.5194/wes-8-1821-2023 But since its for rollers, it is ok to omit it.
R: According to this comment, we have enhanced the literature review, and included the references.

4- The literature list is a bit messy, with fist and last names appearing in different orders and some references being cited without year in the text. Kindly see to its consistence without me having to go through every single reference.

R: We have checked and reviewed all references of the literature.

Individual comments:

1- Line 1: How is the growth extraordinary and in comparison to what?
R: In this sentence we wanted to highlight the significant growth of the wind energy sector compared to the general industry. It is true that when we say that something is extraordinary, we should also point out in front of what it is extraordinary. We have decided to rewrite the sentence:

"Wind industry has remarkably grown over the last decades compared to the general industry...'

2- Line 23: 'extreme operating conditions' is a standing term in IEC61400. Please consider a rephrasing here, because it seems misleading.
R: Following this comment, we have rephrase the term base on IEC61400

3- Line 25: In the given references, I doubt there is a methodology for design against fretting with a safety factor. Either split your statement into two considerations of RCF and wear or be more precise in the sentence.
R: we consider that the sentence can be better understood by exchanging the word 'with' for the word 'and/or' in this way:
"Although new calculation procedures and design guidelines have been proposed for the analysis of bearing raceway reliability in terms of rolling contact fatigue and fretting damage, these techniques rely on simplified models and/or high safety factors which generally result in conservative calculations, and low design optimization (Harris, et al., 2009.) (Portugal, et al., 2017) (Houpert, 1999) (Schwack, et al., 2016) (Heras, et al., 2017) (Olave, et al., 2018) (Lopez, et al., 2019)"

4- Line 28: It would be 'extensive usage' instead of 'extensively'. But, how do you know this was used extensively? And what would be a normal usage? Also, the reference Stammler et al. is unclear.
R: Following this comment, we have rephrased:
"This fact has promoted the usage of experiments to test the reliability of the wind turbine components under their different failure modes".

5- Line 30: Please consider singular and plural in this sentence.
R: We agree that verb should be in plural. It has been corrected.

6- Line 31: Please look up the Oxford comma and use it throughout the document.
R: We agree that we have not used the appropriate Oxford comma in some sentences, therefore, we have reviewed the document for errors of this type

7- Line 32: Please look up the meanings of the words continuous and continual and decide which one to use here.

R: We have reviewed the use of continuous and continuous. As far as we can understand, continual refers to things come and go, like arguments or rain, while continuous is nonstop action. When we refer to wind turbines, we want to emphasize the fact that larger and larger components are being designed and built, and that it seems that this will continue to be the case in the future, or at least not in the future.

8- Line 33: Under this scenario, ... I could not possibly put that into a more complicated sentence. If you wanted to say: "This raises the need for design methodologies and scaled testing approaches whose results are valid for real scale applications." Could I suggest to simply say it?

R: We agree that we have complicated more than necessary. Taking your suggestion we have modify the text.

9- Line 34: Please consider singular and plural in this sentence. Also, I think they do not aim to link hub and blade, they simply link them. It is not like they strive to achieve it and oftentimes fail.
R: You are right from the strictest point of view of the meaning of words, so we take your comment into account and modify it by deleting the word aim, since as you mention it is not an objective but a consequence.
Now the text looks like:

"They link the blades to the hub and allow pitch control to…"

10- Line 35: 'Increase turbine efficiency': Please be more precise. They want to maximize lift of the airfoils. Also, the bearings are not exposed to oscillatory movements, they simply do them.
R: We agree with the comment and have amended the text in both sentences:
"in the optimal position to maximize lift of the airfoils and…"
And here:
"… these bearings usually oscillate, and…"

11- Line 37: Did you mean 'movement' instead of 'moment'? Also: lower than what?
R: As you mention the correct word is movement, and we modify the comparative adjective lower by the adjective low enough adding the reference to Harris' previous work where he points out a critical angle for the development of fretting in pitch bearings.
"… the movement can be low enough (Harris, Rumbarger, & Butterfield, 2009.) to result in rotational…"

12- Line 41: The references fall arguably short of the available ones in this field. Please look up works by Bartschat, Bayer, Behnke, Stammler.
R: We agree that we were missing some references, we have added them.

13- Line 44: I most certainly did not propose to use something called 'endurance runs' as a lubrication or protection run.
R: We had certainly misunderstood some points in the paper (Stammler, Poll, & Reuter, 2019), and the name used to refer to the premeditated use of the control movements. We have modified the text:

"Therefore, the premeditated use of the control movements, called protection runs, have been experimentally studied to observe their effects on the damage to avoid damage (Stammler, Poll, & Reuter, 2019)."

14- Line 45: Please remove the superfluous comma here.
R: Done

15- Figure 1: Please give a full nomenclatoric description in the text

R: We have reviewed the description of the figure and have added all the references to the nomenclature present in the diagram: "... Q is the normal load over a contact, α is the contact angle, $r_c$ is the contact deformed radius, r is the distance from a contact point to the rolling axis, r´ is the effective rolling radius, W is the local rigid slip, T is the tangential stresses, S is the local sliding and $E_r$ is the rolling energy."

16- Figure 2: As mentioned above, please follow the nomenclature of DG03
R: we have updated the variables "$r_p$" by "$r_w$" according to the DG03, and I have deleted variables b, and h as they were not used in the formulation.

17- Figure 2: You drive me especially up the wall when you insist on denoting the pitch diameter as main diameter with the sign r_pw.
R: We apologize for calling radio pitch this way. We agree that it was a mistake. We have modified it in the complete paper.

18- Figure 3: It is unclear how you calculated the pitch bearing rate.
R: Again we agree that the reference is wrong. We have added the reference to the literature where we got the graph

19- Section 2.3: This paper uses a one-third model which does not follow the recommendations of the new DG03 and also neglects results presented by Daniel Becker et al. of thyssenkrupp rotheerde. The results, which indicate the critical zones around the circumference of the bearing, are most likely influenced by this. For the purpose of the study, this shortcoming is acceptable, but needs to be pointed out more clear both in this section, i.e. by stating that cross-influences between the blade roots are thus neglected, and in the conclusions.
R: We agree that it is an effect that can significantly affect the precision of the results. Following your advice we have added the following text:
"It is important to mention that the results in the areas around the circumference may be affected by this assumption by not considering the cross effects at the root of the blade"

20- Section 2.3: The model of the blade is really just the model of a circular tube that neglects both blade geometry and material properties (except for the first few m of a blade root). This again is acceptable but needs to be pointed out clearly.
R: Again we agree that it is an effect that can significantly affect the precision of the results. Following your advice we have added the following text:
"... in the absence of a realistic blade design to carry out the calculation, a laminate circular tube with constant section equal to the bearing ring has been incorporated to apply a geometric offset where the force can be applied..."

21- Line 136: "Mesh is covered with second order elements.." This leads to the question which type of elements the mesh itself consists of, and why did you cover it with additional elements?
R: At this point I think we have expressed ourselves incorrectly. We do not mean that the mesh has been covered but that the entire mesh consists of second order elements.

"The entire mesh is composed of second order elements and..."

22- Section 2.3: Please describe how exactly you connected the nonlinear springs to the raceway (which element type and which size) and also how you evaluate the contact angle. How do you account for a deformation of both bearings rings together, i.e. a tilting of the hub interface? How exactly did you apply and verify the bolt pretension of 608 kN?
R: We certainly haven't explained this point carefully, we just talked about using MPC contacts. We have actually generated partitions of rectangular faces along the raceway of dimensions a_max, b_max. In these partitions, an ordered mesh of 2x8 elements is generated and this is fixed to the end of the linear element at its opposite point as seen in the figure 4c. We have added the text:
"To get this aim, rectangular partitions along the bearing raceway of dimensions a_max, b_max. In these partitions, an ordered mesh of 2x8 elements is generated and this is fixed to the end of the linear element at its opposite point as seen in Figure 4c"
In the case of bolt preloading, it is done through the ANSYS bolt pretension utility. We have added extra text in the manuscript better explain these points.

23- I adore Figure 4, it is a very nice way to present an FE model.
R: Thank you very much for the complement.

24- Table 2: The notation for the selected bending moments seems a bit misleading to me. For load case 4, as an example, did the authors use the maximum Mx times (-1) as the notation would suggest or did they use the minimum Mx? If they used the latter, might I suggest to use Mx,max and Mx,min as notation instead?
R: We certainly use the minimum. Thanks again for considering the detail. We have updated the entire table.

25- Section 2.4 und Section 2.7 mention Appendix A and B, but these seem are not in the pdf pf the preprint. If this is an issue of WES editoring, kindly show me how to view these appendices.
R: We have included Appendix in the manuscript.

26- Equation 3 – 5 : The nomenclature in the text describing the equations is incomplete, variables N_ij, S_ij, Poisson ratio and r_c and s are missing. Note also the previous remark on DG03 nomenclature.
R: We reviewed the nomenclature and now it reads as follows:
"... $v$ is the Poisson´s ratio, $S_{ijk}$ is the sum of the contact curvatures of the groove curvature at row $j$ and raceway $k$, $N_{ijk}$ is the normal load at the ball $i$, row $j$ and contact $k$...".

27- Line 158: Please cite Hertz' work as follows:

Hertz H. Über die Berührung fester elastische Körper und über die Härte. (On the contact of rigid elastic solids and on hardness). *Verhandlungen des Vereins zur Beförderung des Gewerbefleisses*, Leipzig, Nov. 1882. (For English trans. see Misc. papers by H. Hertz, Jones and Schott, Macmillan, London, 1896).
R: Done, we have rewritten this reference.

28- Section 2.7 head: Please use Oxford comma

R: Done, we have included the Oxford comma.

29- Figure 5: I might have missed it, but it does not seem to be mentioned in the text?

R: It is mentioned in line 159 (old version).

30- Section 2.8 : It remains unclear to me if E_f is calculated in absolutely coordinates on the raceways, i.e. taking account of the contact angle changes, or if you calculated it only as a function of their position in the contact area. It becomes clear with the results section, but maybe you can add some more explanation in here?

R: We agree with both comments, we have not correctly explained how the EF variables and their density accumulate, nor have we clarified the difference between TDFE and TFE. Therefore, we have added the following paragraph with the intention of clarifying both points:

"Then, both the dissipated energy, E_F, and its density, ρ_F, are instantaneous variables in local coordinate systems. To achieve the cumulative evolution of damage, these are translated into cumulative variables, Accumulated friction energy, TFE, and accumulated energy density, TDFE, on the global coordinate system. Thus, considering the change in position due to the variation of the contact angle.".

$$TFE = \int_0^t E_\mathrm{F}(t) \cdot \mathrm{d}t = \int_0^t \rho_\mathrm{F}(t) \cdot \mathrm{d}t \tag{1}$$

$$TDFE = \max\left(\int_0^t \rho_F(t) \cdot dt\right) \tag{2}$$

31- Section 3.1 presents results for TFE, which previously in Section 2.7 is denoted as E_F – for the sake of consistency I would suggest to keep E_F after section 2.7. Similar for DFE.

R: We answered together with the question 30.

32- Figure 6: This is a beautiful Figure. Can I suggest adding a table of the maximum values for each of the cells of the Figure?

R: Taking your advice we have added the maximum values on the same table at the location of each maximum.

33- Line 250: "The distribution of the mean load with highest values located at D1 at 150°, and D2 at 315° seems to reveal the presence of a permanent tilting moment of similar components $M_\square$ and $M_\square$ that might be attributed to the mean load exerted by wind."

I think this is too cautiously put as this is an obvious relation? M_y is caused by thrust which is always there at power production, and M_x by blade weight.

R: We also agree with this comment, we have modified the sentence and now it reads as follows:

"The distribution of the mean load with highest values located at D1 at 150º, and D2 at 315º shows the effect of a permanent tilting moment of similar components M_x and M_y attributed to the mean load exerted by wind".

34- Line 303: TFE instead of TEF? (Typo)
R: We have corrected the mistake.

35- Line 371 develop instead of developed
R: We also have corrected the mistake.

36- In the conclusions, I strongly recommend mentioning again the absence of lubricants in your models, especially in the 80 minutes you gave as an exemplary critical time.
R: As mentioned before, we have finally decided to remove this calculation from the paper.

Finally, and again, the authors thank the reviewer for his time and effort in the revision that without any doubt it has improved the work

**ANSWERS TO RC2**:

First of all, we would like to thank the reviewer for the time taken in reviewing the paper, especially in view of the numerous comments and their depth, which will undoubtedly help us improve the paper.

We have tried to cover and dedicate time and care to every comment, and we hope that you find a satisfactory response in them.

In this paper, the authors present a methodology for prediction of the initiation of wear damage in pitch bearings, both for the case of oscillatory movements and rotatory standstill under varying loads. In general, I found myself wishing for more discussion of the results – especially around Figures 8 to 13 and what they really mean.

For example, even the matter of "initiation of wear damage" I don't believe is really explained anywhere – can the authors provide an example (a picture) of what this practically looks like? I was left with the feeling that, well, after you install a pitch bearing and loads are applied in standstill (or small oscillation angle) cases, it quickly develops "a bit" of fretting wear. Too bad for the bearing and all, but isn't that almost to be expected? How much does it matter? And what about the part "in the absence of lubricant"? Is that for real, if so, isn't this completely impractical?

It would have been interesting (in this work) or will be interesting (in another follow-on one) to apply the methodology to SCADA data from operational wind plants to see if there is correlation between the model and pitch bearing replacement records. I feel this even more strongly after reading the statement "In previous work (Cubillas, et al., 2022), the authors developed *and validated* a complete methodology to predict fretting damage in static angular bearing subject to variable loads. In this work, this methodology is extended and adapted to the 4PCB bearing problem." I agree that it is a journal-worthy contribution to extend this model to 4PCB bearing types used as pitch bearings, but boy what will it take to make that next step of validation? That is truly a worthy endeavor. However, I realize this is difficult without a significant amount of 1s time series data from a number of turbines and accompanying inspection or maintenance records. Still, it is in the realm of the possible and I recommend mentioning it as future work in the Conclusions section. The Conclusions point to a "critical region between 270 to 315 of the inner ring and at the first row" – do we know in pitch bearings that have been removed for being damaged, if this is the location? Or is that entirely dependent on the blade root, bearing, and hub models used for the reference wind turbine? If so, does it even matter then?

R: From the comments offered at this point we understand that there are 3 points:

1) Comments related to the discussion of the results sections. In this sense we have worked to improve the discussion, and we have rewritten a good part of the final sections trying to obtain deeper conclusions. As you can see, we have also eliminated the critical time calculation section. The reason is the common coments for both reviewers of the lack of modeling of the lubricant. In this sense we have deeply think on the scope of the paper and the validated frame of the method and have concluded that we may have overextended the capacity of the

proposed method, especially to obtain the wear and critical times. We consider that the lack of modeling significantly affects the calculation of critical times and wear. However, and based on the available literature and previous experiments of the authors in lower scales, we believe that the identification of critical zones, the comparison and analysis of the effect of wind and operating conditions are not so affected (in qualitative terms) by the lubricant. For these reasons, we have eliminated the time-critical sections, and we have enhanced the result discussion.

2) Comments related to the importance or severity of fretting. We agree that at first instance, the fretting wear failure mode does not seem critical, and as you say, it is also to be expected. However, sources in our environment have warned of growing concern in the sector, having detected serious damage to bearing tracks without much use, which triggers other more critical failure modes.

3) In reference to the validation of the results. We agree that the results have not been contrasted with realistic cases as unfortunately we do not have images available for publication. In this direction we would like to continue working to validate the methodology on a real scale, and the proposal to work with the SCADA case study seems very interesting. We discarded this point for the current paper, but we find it very interesting, and we will take it into account. In line with your suggestions, we agree that we should mention the validated status and clarify next steps in the conclusions. That is why we have added the following text:

"The results obtained in this work put light on the fretting damage mechanism in pitch bearings under different conditions of wind and operation. However, there is still a long way to go. Among these tasks, the most obvious would be to validate the proposed methodology on a real scale, with and without lubrication to see the validation framework, as well as the need to implement additional formulations for modeling the effect of grease. Additionally, the formulation should be extended to the prediction of wear over time, making the methodology usable as a prediction tool for design."

Technical comments:

Line 25: I believe this manuscript was likely submitted prior to publication of the revised DG03 pitch and bearing design guide. I recommend that https://www.nrel.gov/docs/fy24osti/89161.pdf be added to the list of citations here since it has been recently published.

R: We have added this citation throughout the entire paper.

Line 26: I believe my fellow reviewer disputed the statement "fretting damage...supported by simplified models with high safety factors...". However, at least for the original DG03 by Harris, it does include calculations and estimation of a fretting corrosion safety factor in Section 7. Therefore, at least for this Harris reference, I support this statement. Maybe it needs to be tailored a bit more, but I think in general it is correct.

R: We have reviewed the original DG03 for the safety factor. As far as we have managed to understand, this guideline points out two critical oscillation angles for the development of fretting corrosion but, as mentioned by reviewer 1, it does not specify that a safety factor is needed. In any case, we do consider that this formula is too simple and does not consider transverse rotation, which is the heart of this paper.

We have rephrased the sentence:

"These techniques are supported by simple methods and/or high safety factors that generally result in conservative calculus, and a low design optimization"

Line 32: I don't understand the sentence "Under this scenario, wind sector claims for the development and validation of new methodologies in multiscale components for the final application to real scale problems (Olave, 2019)." Please revise this sentence. I read it a few times and I really can't even make a guess as to its meaning.

R: We have rewritten this sentence and now it reads as follows:

"This raises the need for design methodologies and scaled testing approaches whose results are valid for real scale applications".

Line 43: I believe this sentence is better stated as "Therefore the premeditated use of the control movements, called endurance runs, have been proposed to avoid damage (Stammler et al xxx)". The citation is also missing the year. Having said that, I believe the term might have been "protection run", but please check it.

R: We agree that both the reference and the term were incorrect. We have modified the text in this way:

"Therefore, the premeditated use of the control movements, called protection runs, have been experimentally studied to observe their effects on the damage to avoid damage (Stammler, Poll, & Reuter, 2019)".

Line 56: This sentence is important, as it describes the paper. However, "to analyse the performance of the damage" is not really clear. I'm also quite puzzled by the parenthetical statement "in the absence of lubricant". Does this really mean that the lubricant is not considered in the model at all? Is that why the apparent 80-minute time to induce fretting damage mentioned in the Conclusions is so low?

R: According to the new scope of the paper, we have rewritten the objective of the paper and we hope it is now clear:

"The main objective of this work is to propose a complete methodology for the analysis of pitch bearing fretting damage, and to analyze the fretting damage under productive and non-productive periods of static pitch control, to determine the critical locations, and assess the worst working conditions"

Regarding to the "absence of lubrication" we have written the following in the introduction:

"It is important to mention that the present analysis does not consider the effect of the lubricant. However, considering the boundary lubrication regime condition of this case of study, as well as previous work where an adequate correlation between friction energy density intensity and damage has been found, it is considered that the proposed method generates a reliable framework for the prediction of the most critical areas where damage can occur, as well as providing a basis for comparison between different wind speeds and operating conditions."

Certainly, the 80 minutes was a result of the lack of consideration of the lubricant and therefore we have eliminated that section.

Lines 60 and 384: I believe the citation and bibliographical entry for IEC 61400-3-1 should be updated, in terms of both the title and publication date. Please see https://webstore.iec.ch/en/publication/29360. Also, as written, the sentence implies the wear energy-based model is contained in IEC 61400-3-1. I am not an expert in that document, but is that correct? I think more likely the intent is that the DLCs were derived in accordance with IEC 61400-3-1. If so, I recommend this sentence simply be "...different design load cases (DLC) of normal production (DLC 1.2) and non-productive conditions periods (DLC 6.4) from IEC 61400-3-1 are evaluated through a wear energy-based model." Really, I don't think a citation to or bibliographical entry for IEC 61400-3-1 are necessary, as the reader has all the information needed to find the document by just mentioning it in the text. If the authors would like to keep it, then no problem, but the entry needs to be corrected.

R: We agree that we did not describe this point correctly, and took your advice and proposed text as a better wording:

"... different design load cases (DLC) of normal production (DLC 1.2) and non-productive conditions periods (DLC 6.4) from IEC 61400-3-1 are evaluated through a wear energy-based model".

Likewise, we agree that the reference can be removed.

Line 91: As written, this first paragraph in Section 2 ends abruptly with "Finally, the fretting damage and the probability of damage initiation are calculated." I would recommend adding a simple sentence here that serves as a transition "Each of these steps are described in the following subsections." Or something of a similar nature.

Once again, we agree that, seen in perspective, a transition text should be added. For this reason, we also include the proposed text.

Line 131: I will admit I'm not an ANSYS user. Is it true that ANSYS considers the "geometry" of the bolts? Line 138 later states "bolts are replaced by linear elements".

R: ANSYS would have the capacity to include realistic screw geometries, however, this entails a high computational cost, especially when such a high number of screws are considered. For this reason, and with the aim of simplifying the calculation, the bolts are replaced by beam elements with an equivalent associated section according to the VDI.

Line 145: What is an "MPC contact"? Sounds like an ANSYS thing but should be defined if it is an acronym.

MPC contacts are multi point constraint contact that adds internal constraint equations to tie the displacements of the contacting surfaces. I would say that are very well-known contacts and behaviors for ANSYS users, but we agree that for no users can be misled. We have added the acronym to the sentence.

Lines 156 and 185: Both Appendices are missing from the pre-print.

R: We have included both appendices to the revised preprint.

Figure 6 and discussion: This is quite a figure that takes some time to digest. Aside from the discussion of it, I want to be sure I understand it more generally. At first, I assumed

each circle represented a ball-raceway contact. Is that correct? I didn't count, but are there 121 of them? Or is each point just a discrete location? Additionally, and it could be a preference thing, but the use of different color scales for the different DLCs made it a bit hard to understand which DLC (1.2 or 6.4) might be more damaging? Is that a conclusion that can even be made?

R: Each point represents the maximum value of two adjacent balls (since the number of balls was too high and the difference between two adjacent balls was low), as mentioned in the text:

"... and attending to the high number of contacting points and the similarity between adjacent locations, the results correspond to the mean value of two adjoining balls with the objective of facilitate the presentation of the data."

Regarding the scales, there is too high a difference between the damage scales of DLC1.2 (around 1.1J) and DLC 6.4 (with a maximum of 0.06J). For this reason, it was decided to use a different scale. However, we consider that you are right in that this has not been commented on in the text, which is why the following text has been added:

"Furthermore, considering the large difference between the damage values generated in production and non-productive times, it has been decided to use different scales."

Figure 7 and discussion: The x-axis I believe to be a portion of the circumference, as described in the text. However, I'm not sure I understand what the y-axis is. Is what I'm looking at analogous to the contact patch for a ball-raceway contact? If so, what is the top and bottom? That is, where are these relative to the raceway edge or the contact angle? As you can see, not being intimately familiar with some of these new metrics – TFE and DFE – the article could benefit from some additional simple text describing the nature of the plots and axis labeling and things to orient the reader a bit more.

R: Figure 7 shows the density of the friction energy, DFE, along the different bearing raceways, being the axis x, the longitudinal direction of the raceway and the axis y the transversal direction, according to the local coordinate system of Figure 5.
Here I leave you another image that we were tempted to use but that we discarded because it took up too much space. It is true that having worked on this topic for a long time we are too used to it and for an external reader it may not be difficult.

[Figure]

We have added extra axis and we have referred them in the text to the Figure 5 where axis are clear.

Figure 8 and discussion: Although the article doesn't say it explicitly, but is the conclusion that these wind speed and DLC cases are much less probable to initiate fretting then the cases in Figure 6? That is my quick takeaway, but I am not sure if that is correct. Regardless, can some discussion be added comparing Figures 6 and 8?

R: Figure 8 shows the effect of the wind, and it only shows the accumulated damage for a 10-min time series of each of the wind intensities. That is to say, the sum of all of them would give the result observed in figure 6. The maximum dissipated energy in this case is 0.6, which would tell us that the most damaging wind (7m/s) produces (1.1-0.6)/1.1 = 0.45

of the total. But I don't see much force in this conclusion, I think it can be expected. What we do consider interesting is that despite the analogous growing tendency on both indicators with the growing wind speed, the tendency is significantly different: while the TFE has an exponential growth, the MDFE shows a logarithmic tendency. As we indicate in the text.

Figure 10 caption: The text refers to the "ball located in D2R1 at 275" while the caption refers to "ball 105". I think I prefer the former, please be consistent though. In the figure legend, I believe "lineal" should be "linear"? I believe these are just simple linear fits to the results from the DLCs themselves, honestly, I don't know that these are necessary – one can see the trend for each is linear.

R: We completely agree that there is no consistency, we also consider that the most appropriate way to refer to the ball is D2R1 at 275. That is why we have updated the caption.

Generally speaking, once we get to Figures 8 to 12 especially, the figures are accompanied by relatively little text and I found myself looking at them and hoping for more discussion. I recommend revisiting these and encourage thinking more about the main messages of each.

Figure 12 and 13: The x-axis for the a) wind speed plot goes from 0 to 300...and without units. Is this right?

R: Sections have been removed from this manuscript.

Here especially in Figure 12, I think this is the real "heart" of the paper, yet the discussion is barely 3 sentences. For example, what do we make of this fast transition in probability for DLC1.2? I am really confused by the x-axis here as well.

R: Sections have been removed from this manuscript.

How do these amounts of time spent in any particular DLC translate to amount of total operational time? For example, for a typical site wind speed class, I think you can assume a distribution of wind speeds. So can one "add up" these probabilities and project how many years it might take to reach 100%? Is it 1, or 5, or 10, or 20 (typical design life) or 100? I have no idea, but it would be nice to get a sense of that.

R: Sections have been removed from this manuscript.

Is this what Figure 13 is getting at? I will admit I had a hard time understanding what Figure 13 was about, as the x-axis time periods are so small – minutes and days. The Conclusions get to this with "quick and concerning mechanism of damage in productive periods that can developed damage after 80 minutes in the worst scenario". Aside from the fact that this is a little hard to believe, how was that concluded...from Figures 12 or 13.

R: Sections have been removed from this manuscript.

Where exactly? This conclusion should be highlighted and explained in the discussion of the figures, as is, it feels like it comes out of the blue in the Conclusions.

R: Sections have been removed from this manuscript.

Minor grammatical comments:

R: All these grammatical comments were address.

Line 22: I believe "technics" should be "techniques".

R: Corrected

Line 27: "what" should be "that".

R: Corrected

Line 28: "extensively usage" should be "extensive use".

R: Corrected

Line 34: I believe "and it aim is to link" is better stated as "that connect".

R: Corrected

Line 37: I believe "moment can be lower" should be "movement can be low".

R: Corrected

Line 49: "what result" should be "that result".

R: Corrected

Line 52: "authors has" should be "authors have".

R: Corrected

Line 72: "what allow create" should probably be "that creates" or "that allows creation of"

R: Corrected

Table 1: "heigh" should be "height".

R: Corrected

Figure 3 caption: Includes [10], but I am assuming this should be a citation to something else, as the citations are not numbered?

R: Corrected

Line 121: "As describe" should be "As described"

R: Corrected

Line 123: "is not operational. To manage this inconvenient" is probably better stated as "is not practical. To manage this inconvenience".

R: Corrected

Line 124: "what allows to create" should be "that allows for creation of".

R: Corrected

Line 250: the degree symbols here don't actually appear to be degree symbols.

R: Corrected

Line 295: some commas are missing and damage is spelled incorrectly.

R: Corrected

R: Corrected

---

## Author Response (AR2)

I appreciate the thoroughness of the authors' responses and changes.

R: We are very grateful again for the degree of involvement in the revision, which has undoubtedly allowed us to improve the manuscript.

Having said that, I still recommend the following:

Introduction:

- I see the existing reference to Harris 2009 and the new reference to Stammler et al 2024 in lines 27-28. Regarding the matter of a fretting corrosion factor and aside from the critical oscillation angles, Table 16 in Section 7 of Harris 2009 does provide a recommendation for maximum stresses for minimizing fretting corrosion. When compared to the operational stresses, this does provide a safety factor, so I so support the use of the phrase "...or high safety factors..." here. This Section in Harris 2009 also speaks to the importance of grease additives with respect to boundary lubrication, coincidentally. I only mentioned this here for our own understanding – I am not suggesting that anything be changed in the manuscript in this respect.

  R: We agree that although the existence of a safety factor is not explicitly specified in Harris, a comparison with respect to the angle of rotation can provide it. We also appreciate this clarification for our own understanding.

- I appreciate the corresponding changes with respect to the main objective of the paper in lines 59-61. However, regarding the revisions I think calling this a "complete methodology" might be a little overstated. I might suggest revising these lines to be a bit more direct "The objective of this work is to propose a methodology to analyze the relative contributions of productive and non-productive periods of wind turbine operation to pitch bearing fretting damage, including identification of the most likely location of damage on each raceway."

  R: Again, we agree that after the discussion in the previous review of the actual capabilities of the methodology, referring to it as complete is a bit overstated. Since the proposed phrase fits perfectly with the work, we have decided to take it.

- Line 64: as written, this sentence still implies the wear-based energy model is in IEC 61400-3-1. Please change to "IEC 61400-3-1 (61400-3-1, 2019) are evaluated through a wear energy-based model." and update the citation or "IEC 61400-3-1 are evaluated through a wear energy-based model." and delete the citation.

  R: We have revised this point and deleted the reference

- I think the added description in lines 65-70 is very useful, but I might still recommend further tweaking it a bit to be more explicit. The old adage that "all models are wrong, but some are useful" applies here I think, so as always it is very important to state the conditions under which the model applies and for what it can be used to determine. I recommend something like the following "In this work, the damage evaluation has been addressed using energy-based wear models that

have previously demonstrated adequate correlation to damage (Brinji et al., 2020), (Schwack et al., 2018), (Cubillas et al., 2022). Although the present analysis does not consider the effect of the lubricant, pitch bearings are likely to be in boundary lubrication conditions when the pitch angle is not changing in non-operative periods and even in some low wind speed operations in which the model is valid. Therefore, the authors believe that the proposed method provides a reliable framework for the prediction of the most critical areas on the raceway where fretting damage can occur as well as a basis for comparison between the severity of different wind speeds and operating conditions." I think this is the best (clearest) statement of the paper's hypothesis Having said that, I'm not enough of an expert in fretting corrosion to weigh in on its validity. I would be especially interested in my fellow reviewers' opinion in this respect.

R: We agree with this point as well, we like the proposed sentence, and we take it for the manuscript.

Methodology

- Caption of Figure 2 and Table 1. The variable r_pw is specified as the "Bearing pitch diameter", but the variable name implies a radius. The figure itself could be either, although 3610 mm would seem to be a diameter for a turbine of this size. If it truly is a diameter, I recommend D_pw to align with typical usage.

R: The error is in the nomenclature, it should be d_pw, we have corrected it.

Results and Discussion

- Figure 7 I will admit is better but maybe a but confusing. In Figure 5 it looks like x' is vertical (wrt the page itself) and y' is horizontal (also wrt the page). The discarded figure you show in your author's response has a similar orientation. However, is Figure 7 "flipped" as shown, in which x is horizontal and y is vertical? Or are these mislabeled and should also be x' and y'? To be honest, I rather like your discarded Figures as they are much easier to interpret – in an electronic journal I wouldn't sweat the extra space.

R: Yes, you are totally right, x'/y' direction were flipped. According to this comment, we have added the discarded picture to make the interpretation easier, and we have corrected the x', y' axis.

- Line 310: the statement "Therefore, it appears that locking the rotor at non-productive times DLC6.4 could help to limit the damage" could be met with some amount of resistance. Although from this analysis this might help the pitch bearing, I believe many would say it could harm the main bearing, gearbox teeth and bearings, and generator bearings. Additionally, compared to DLC 1.2 isn't the fretting in DLC 6.4 low (maybe even 3 orders of magnitude lower in Figure 8)? At least, that is my takeaway. So, practically speaking, even if you did lock the rotor, would it matter? The last Conclusion also reinforces my point here.

R: Considering this comment, we have decided to delete this sentence.

- Lines 327-331: I appreciate the addition here with respect to new work, however, I recommend it be moved to the Conclusions as this generally applies to the whole thing and not just what can be concluded from Figure 9 which immediately precedes it.

  R: We fully agree that this text would work better in the conclusions section. We have moved it.

Minor grammatical comments:

- Line 26: I would add "..of the pitch bearing raceway..." here.

  R: Corrected

- Line 77: "what allows to create" is better phrased as "that allows creation of" or "that determines"

  R: Corrected

- Line 304: "dagame" is still mis-spelled.

  R: Corrected

Again we would like to thank the reviewer for his time and comments.